# Comparison of Prediction Models for Mortality Related to Injuries from Road Traffic Accidents after Correcting for Undersampling

**DOI:** 10.3390/ijerph18115604

**Published:** 2021-05-24

**Authors:** Yookyung Boo, Youngjin Choi

**Affiliations:** 1Department of Health Administration, Dankook University, Cheonan 31116, Korea; dkykb0926@dankook.ac.kr; 2Department of Healthcare Management, Eulji University, Seongnam 13135, Korea

**Keywords:** road traffic injury, mortality prediction, classification technique, national hospital discharge dataset, imbalance dataset

## Abstract

In this study, four models—logistic regression (LR), random forest (RF), linear support vector machine (SVM), and radial basis function (RBF)-SVM—were compared for their accuracy in determining mortality caused by road traffic injuries. They were tested using five years of national-level data from the Korea Disease Control and Prevention Agency’s (KDCA) National Hospital Discharge In-Depth Survey (2013 through to 2017). Model performance was measured for accuracy, precision, recall, F1 score, and Brier score metrics using classification analysis that included characteristics of patients, accidents, injuries, and illnesses. Due to the number of variables and differing units, the rates of survival and mortality related to road traffic accidents were imbalanced, so the data was corrected and standardized before the classification models’ performances were compared. Using the importance analysis, the main diagnosis, the type of injury, the site of the injury, the type of injury, the operation status, the type of accident, the role at the time of the accident, and the sex were selected as the analysis factors. The biggest contributing factor was the role in the accident, which is the driver, and the major sites of the injuries were head injuries and deep injuries. Using selected factors, comparisons of the classification performance of each model indicated RBF-SVM and RF models were superior to the others. Of the SVM models, the RBF kernel model was superior to the linear kernel model; it can be inferred that the performance of the high-dimensional transformed RBF model is superior when the dimension is complex because of the use of multiple variables. The findings suggest there are limitations to analyses involving imbalanced, multidimensional original data, such as data on road traffic mortality. Thus, analyses must be performed after imbalances are corrected.

## 1. Introduction

Road traffic accidents are the leading cause of accidental casualties. As they are increasingly contributing to physical injuries, deaths, and disabilities after treatment, road traffic accidents are being recognized as a major public health problem [1]. Road traffic injuries (RTI) are injuries caused by road traffic accidents involving motor vehicles. They include between-vehicle collisions, vehicle-pedestrian collisions, and vehicle collisions with animals or fixed objects [2]. Injuries due to road traffic accidents are a major public health problem, especially in Organization for Economic Cooperation and Development (OECD) member countries where they caused more than 100,000 premature deaths in 2013. The direct and indirect financial costs of traffic accidents are reported to be substantial, accounting for 1% to 3% of a country’s annual GDP [3]. Furthermore, in addition to causing human suffering, road traffic injuries can place a significant financial burden on victims and their families due to treatment costs, deaths, and the reduction in productivity caused by disabilities [4], and nearly three-quarters of the deaths occur in men.

In the past, injuries from vehicle accidents were perceived as unavoidable events that occurred without notice. However, perceptions have been evolving in recent years. Changing lifestyle habits and new national strategies for managing injury occurrence as a public health problem, not only a safety issue, have led to the view that traffic accident-related injuries are preventable. As a result, it has been recognized that similar to many illnesses, it is possible to prevent and reduce the incidence of external injuries and related deaths with appropriate management strategies [5]. Therefore, it is important, from a medical perspective, to establish prevention, treatment, and rehabilitation guidelines by determining the scope of injuries and evaluating their causes or epidemiological characteristics [6].

As deaths from road traffic accidents result in significant social and personal loss, such losses must be minimized by strengthening the system to better prevent and manage injuries [7]. Classification is a data mining technique that can be used in predicting mortality resulting from road traffic accidents, as sufficient training data must be obtained to construct a model. Accident datasets are useful from a transportation perspective; from a medical perspective, they are a source for information about external causes of injury. In Korea, the National Hospital Discharge Survey has been conducted annually by the Korea Disease Control and Prevention Agency in Osong (KDCA) since 2005 to produce health statistics and determine the scope of injuries at the national level regarding discharged patients in health care facilities. This survey is similar to the National Hospital Care Survey, which is conducted by the National Center for Health Statistics, a component of the United States Centers for Disease Control and Prevention (CDC) [8]. As it is a national survey, the sample size and reliability of the data from the National Hospital Discharge Survey are adequate. The Korea National Hospital Discharge In-depth Injury Survey is a national survey, and the reliability of the data and the amount of sample are sufficient, but it is unbalanced data with a large number of survivors and a small number of deaths to analyze the mortality caused by a traffic accident, thus, there are limits to using the unprocessed data sets. Additionally, it takes a long time to perform analyses owing to the large size of the data.

Survivors and the deceased are distinguished during the classification process; when there is a significant difference in the number of observations between the two classes, the data are said to be imbalanced. This difference makes modeling difficult because when such imbalanced data are used to perform classification analysis, the data class with the largest number of observations dominates the classifier creation process. However, the class with the smaller quantity of data also provides significant information. Incident severity data sets are generally unbalanced, and there are more non-fatal classes than fatal classes. To handle such unbalanced data sets, over-sampling and under-sampling methods are often applied [9]. As one of the over-sampling methods, SMOTE (Synthetic Minority Oversampling Technique) is not only applicable to various problem solving, but is widely used in fields such as multi-label classification, progressive supervised learning, and semi-supervised learning due to the simplicity of the procedure [10,11].

To solve the classification problem, the method of analysis must first be selected. Traffic accident research has been conducted on the factors that cause accidents and to estimate the severity of injury caused by accidents. To evaluate the causes of traffic accidents, contributing factors were classified into human, automobile, and environmental factors, as the factors that must be controlled for the prevention of accidents [12,13]. There are studies to estimate the relationship between the level of injury caused by an accident and the severity of the injury from a medical aspect. In these cases, probability models have been mainly used for traffic accident research, and logistic regression has been used in binary studies, especially mortality [14,15]. Wei and Chiu (2002) used decision trees to perform their analysis, while Coussement and Van den Poel (2008) used and compared techniques such as support vector machines, random forests, and logistic regression [16,17]. Lastly, Mozer et al. (2000) utilized techniques such as logistic regression and decision trees in their study [18]. As probabilistic models start with clear assumptions about the structure of the model, statistical biases or erroneous results can be included in cases where such assumptions are violated [19,20]. In order to overcome the limitations of statistical models, a non-probabilistic data mining approach has been widely used recently, and among them, clustering [21,22], classification, and regression tree methods [23,24,25] have been adopted. In addition, among these non-parametric models, SVM was developed as a method to improve regression and classification problems and is used in the study on the severity of injuries in traffic accidents [14,26]. Especially using SVMs, studies were focused on investigating methods to increase its accuracy using radial basis function (RBF) [27]. The most commonly used classification analysis methods are those that employ decision trees, logistic regression, and SVM. After the model is chosen, procedures must be selected to deal with imbalanced data, and finally, it is necessary to consider what measures will be used to validate the models [28].

This study used data from the KDCA’s National Hospital Discharge In-depth Injury Survey to determine mortality related to external injuries caused by road traffic accidents. To predict the mortality of hospitalized patients who were injured in road traffic accidents, this study used the characteristics of patients, accidents, injuries, and illnesses to perform classification analysis and evaluate the performance of each model. In particular, as the sample included a number of variables with different units and the rates of survival and mortality related to road traffic accidents were imbalanced, this study corrected and standardized the imbalanced data before comparing the performance of the classification models.

## 2. Materials and Methods

### 2.1. Research Sample

This study used five years of data from the KDCA’s National Hospital Discharge In-depth Injury Survey, which was reported between 2013 and 2017. All personally identifiable information was removed from the data before it was disclosed to the researchers. The researchers of this study requested the above data from the KDCA for research purposes and were provided the data after obtaining the agency’s approval. The survey population of the National Hospital Discharge In-Depth Survey was defined as all patients who were discharged from general hospitals that have 100 or more beds. The data excluded facilities such as long-term care hospitals, geriatric hospitals, military hospitals, and single-specialty hospitals with 100 or more beds. The hospitals were arranged based on the number of beds using a stratified two-stage cluster sampling method according to the number of hospital beds. Then, 170 hospitals were extracted using the Neyman allocation method, and 9% of the patients who had been discharged from those hospitals after receiving inpatient care were randomly selected to compose a sample [29]. The KDCA used in this study was panel data, and the patient’s personal information was de-identified during the collection process. Our study was approved by the Institutional Review Board of Dankook University (DKU 2021-04-019). The review board waived the requirement for informed consent due to the retrospective design of the study.

The survey items consisted of information related to health care facilities, demographics and geographics, patients’ hospital visits, and diseases and treatments. Injuries were investigated further using injury codes and codes for external causes of injury as well as in-depth, injury-related information. The principal diagnoses of patients were collected using the codes in the Seventh Revision of the Korean Classification of Diseases, the Korean version of the World Health Organization’s Tenth Revision of the International Classification of Diseases (ICD-10). Volume 3 of the International Classification of Diseases, Ninth Revision, Clinical Modification (ICD-9-CM Volume 3) was used to assign codes to principal surgical operations and other treatments.

From the National Hospital Discharge Survey sample, patients with unintentional injuries in which the mechanism of injury was related to traffic accidents were selected as the study sample. Cases of injuries resulting from intentional self-harm, assault, or “other” were excluded, and data cleaning was performed on the first selected sample. The preprocessed dataset was then used for classification analysis. The data cleaning process included the following steps in Figure 1.


If the principal diagnosis does not have an injury code, use the injury code of the other diagnosis as that of the principal diagnosis to match the injury with the external cause of injury code;If the principal or secondary diagnosis does not have an injury code, exclude it as not being an injury (1587 cases);Exclude cases if there is no injury code and there are only complications of medical and surgical treatments (T80–T88);Exclude cases if there is no injury code, and there are only the sequelae of injuries and addictions (T90–T98);If there are multiple types of injuries to one anatomical site, consider it a single site.


Standardized scaling was performed after the motor vehicle accident data were cleaned. The performance of the prediction model is affected by the use of split training and test data. In other words, using data that have been split once can distort a model’s performance. Therefore, a model must be sufficiently verified using datasets that have been split multiple times to validate its performance [30]. The more data splits used, the better the model will converge according to the law of large numbers [31]. Accordingly, as depicted in Figure 2, this study used a dataset generated by randomly splitting the motor vehicle accident data, which were extracted from the database, a total of ten times in order to develop and verify a model.

As the motor vehicle accident data used in this study were imbalanced, a method of oversampling, the synthetic minority oversampling technique (SMOTE), was used to reduce the distortion of the performance measurements that would occur if the original data were to be used. The SMOTE technique generates a linear synthesis pattern between the minority class of interest data and the nearest minority class data [32]. If a simple method of replicating the data from the minority class is used, samples will be generated in specific regions, making it impossible to make proper predictions regarding new minority class data. Thus, this study utilized the SMOTE technique to address this problem. To predict mortality resulting from road traffic accidents by using datasets that had been corrected with the SMOTE technique, this study used logistic regression, random forest, SVM, and RBF-SVM algorithms to perform classification analysis and compare the performance of the models.

### 2.2. Variable Definitions

The vital status of individuals was selected as the dependent variable. Vital statuses were classified as ‘survived’ or ‘died’ based on the survey item regarding treatment outcomes. To identify the factors that affect the vital status, the independent variables were analyzed after they were classified into the following categories: patient characteristics, accident characteristics, injury, and illness characteristics [33]. The patient characteristics were sex and age.

For accident characteristics, this study included the traffic accident mode and the individual’s role in the accident. The following characteristics were selected for injury and illness characteristics: principal diagnosis, patterns of injury, site of injury, the occurrence of a surgical operation, and severity of injury (severe/minor/unknown). The severity of the injury was categorized according to the ICD classification, with superficial injuries classified as minor, and deep injuries classified as severe injuries. The individual’s role in the accident was subdivided and categorized into seven attributes involving: drivers, passengers, and pedestrians. For the mode of traffic accidents, land traffic accidents corresponding to the V00-V89 code of the ICD classification system were categorized into the following nine types: pedestrian, bicycle, motorcycle, three-wheeled motor vehicle, passenger car, pickup truck, van, heavy cargo truck, and bus. Injury sites were classified into the following categories according to the injury site codes: head, neck, spine and back, torso, upper extremities, lower extremities, and “others.” These categories were further divided into subcategories. The aforementioned injury site codes were included in the guidelines for using the original data from the National Hospital Discharge Survey. The types of injury were classified into categories such as single-site or multiple-site injury. Regarding whether a surgical operation had been performed, cases for which there was a date listed for a principal operation were considered cases in which an operation was performed [6,34]. In this study, data from the National Hospital Discharge Survey were used to perform analyses with the Python 3.8.0 managed by the non- profit Python Software Foundation after the data had been cleaned with the Microsoft Excel 2016 (Microsoft Corp., Redmond, WA, USA).

## 3. Results

### 3.1. Sample Characteristics

The sample used in this study consisted of 55,279 individuals with a mean age of 42.72 years. Of these, 32,936 were male, accounting for 59.6% of the total sample. With regard to their role in the accident, 22,358 people (40.4%) were drivers, 9411 people (17.0%) were pedestrians, and 8818 people (16.0%) were passengers. With regard to the site of injury, 20.1% were located in the abdomen and back, 20.0% in the head, and 19.0% in the neck; together these three sites accounted for 60% of the total cases. As for the patterns of injury, sprains and dislocations comprised the highest percentage at 43.5%, fractures accounted for 20.0%, and superficial injuries accounted for 16.3%. Single-site injuries make up the majority of cases at 95.4%, only 21.3% of the hospitalized patients underwent surgery, and 1.2% of the hospitalized patients died after being treated for traffic accident-related injuries. The sample characteristic is shown in Table 1.

### 3.2. Classification Analysis

In this study, the same variables were used to compare the performances of the classification models. Importance analysis was used to analyze the variables of characteristics, such as the characteristics of patients who were hospitalized after traffic accidents, as well as the characteristics of accidents and the characteristics of injuries and illnesses. Only those with variable importance of more than 5% were used in the classification analysis. The variables of sex and level of injury were omitted from importance analysis, while the following variables were selected: principal diagnosis, patterns of injury, site of injury, type of injury, the occurrence of a surgical operation, mode of traffic accident, role in the accident, and age. 

In addition, this study corrected the sample size imbalance between the two groups of patients—those who survived and those who died. When classes of data are highly imbalanced, the model that chooses the majority class will have a greater level of accuracy, which makes it difficult to discern the performance of the models. In other words, even if a model has a high level of accuracy, the recall rate of a class with a small amount of data may decrease rapidly. Such problems of imbalanced data occur because of the difference between the number of data items in classes. Accordingly, this study considered a 98.8% survival rate and 1.2% mortality rate and corrected the imbalanced data. Oversampling and undersampling are methods that can correct such imbalances. In the case of undersampling, there is a significant loss of data, and it is possible that important normal data may be lost. Thus, this study utilized the SMOTE oversampling method [35].

The variables in this study were standardized to minimize the bias that occurs when the variables used in the classification analysis have different units. Methods of standardization in this study included the StandardScaler method, which removes the mean and scales the data to unit variance, as well as the MinMaxScaler method, which rescales data so that all feature values are between 0 and 1. This study compared the distribution of the training, validation, and test datasets after using the two methods of standardization to standardize the data in Figure 3. As a result, it was possible to see the difference in distribution between the training, validation, and test datasets when the MinMaxScaler method was used. Thus, as the distribution of the datasets was better when the StandardScaler method had been used to standardize the data, compared to when the MinMaxScaler method had been used, this study utilized the StandardScaler method.

This study used algorithms commonly used in classification analysis. This included the logistic regression algorithm, the random forest algorithm, which is a decision tree technique, as well as the SVM algorithm. In terms of the SVM algorithm, this study employed both the linear and RBF kernels. Using the RBF kernel method, the given data were mapped into a high-dimensional feature space [36]. After the data are mapped into a high-dimensional space, they can be classified as a linear shape that is not visible in the original dimension. Thus, as diverse types of variables were used in this study, the SVM algorithm was differentiated into linear and RBF kernels. 

After using the four classification models, it was determined that all four models had many outliers (O) that deviated from the quartiles in terms of predicting death (0 on the x-axis). However, the linear and RBF-SVM models had relatively fewer outliers when compared to those of the LR and RF models. In other words, the SVM model had a relatively smaller variance in predicting survival than the other models. 

### 3.3. Model Evaluation

When the accuracy of the models was measured using the original imbalanced data, all models had an accuracy of 98%. In other words, this problem occurred because the data regarding the severity of the injury—a criterion used in the original data—were imbalanced. This signifies a serious problem where the model’s accuracy is at 98% even when the model predicts that the entire sample survived, with the percentage of survivors in the sample being 98.8% [28].

Thus, this study used the SMOTE technique, a method of oversampling, to correct imbalanced data, and it was determined that the accuracy score of the models was between 0.766 and 0.911, with the linear SVM model scoring the lowest and the RBF-SVM model scoring the highest. While accuracy is the most commonly used metric, additional methods have been used to evaluate the models because there was a limit to the accuracy of predicting mortality due to the low frequency of mortality-related cases, even though the data were classified properly. Therefore, this study also used the following metrics: precision, which is the ratio of true positives to all the predicted positives; recall, which indicates the accuracy of a model at locating positive values; F1 score, which is the harmonic mean of the model’s precision and recall in Equation (1), and the Brier score, which is appropriate for binary outcomes. The Brier score measures the accuracy of the binary outcomes that we want to predict. The Brier score for these predictions is shown in Equation (2).
F1 = 2 ∗ (Recall ∗ Precision)/(Recall + Precision)(1)
Brier = mean((y − *p*)^2^) = mean(y × (1 − *p*)^2^ + (1 − y) × p^2^)(2)

Five model evaluation metrics were used to assess the classification models used in this study. The distribution of the evaluation result values ranged from a minimum value of 0.753 to a maximum value of 0.942, indicating that the performance was above a certain level. In terms of accuracy, the models that used logistic regression and linear SVM algorithms had a low accuracy score of 0.76. The model that used the random forest algorithm had an accuracy score of 0.844, and the model that used the RBF-SVM algorithm had the highest score of 0.941. In terms of the F1 score, which is the harmonic mean of the precision and recall, the logistic regression and linear SVM models had low scores of 0.756 and 0.758, respectively. The random forest model had an F1 score of 0.851, and the RBF-SVM model had the highest score of 0.942. According to the evaluation results, the RBF-SVM model was the superior model with regard to all evaluation metrics. The random forest model performed the second best, while the evaluation result values were similar for the logistic regression and linear SVM models in Figure 4. 

When the ROC curves of the models were used to evaluate their performance, the results indicated that the prediction models were valid, because all the curves were located in the upper left of the 45° diagonal line, which extends from the bottom left corner to the top right corner of the graph. Of the models, it was determined that the RBF-SVM model was the best, followed by the RF model as the second best. Furthermore, when the performance of the models was assessed using their precision-recall curves, the RBF-SVM and RF models were found to be superior to the other models in Figure 5. It demonstrates that the non-probability model is superior to the probability model, logistic regression, and an ensemble technique that synthesizes predictions can be applied [37]. Therefore, of the models that classify the mortality caused by road traffic accidents, it can be said that the RBF-SVM model is the best. This shows that RBF-SVM can be superior to linear-SVM in nonlinear cases where the distribution of multiple variables affecting the severity of traffic accidents is difficult to classify as linear [27]. 

## 4. Conclusions

This study used five years of data from the KDCA’s National Hospital Discharge Survey reported between 2013 and 2017 to compare classification models that predict the mortality caused by road traffic accidents. First, the determinant factors of mortality were selected through an importance analysis. The importance analysis result suggests that patterns of principal injury, the role in the accident, and the primary site of injury were identified as major variables. Additionally, internal organ injury was higher in patterns of principal injury, driver’s role in the accident, and head injury, while deep injury at the primary site of injury was identified as the biggest contributing factor.

Utilizing all the features available in the data often helps to improve the performance of the classification model at the most. However, in the medical field, rather than using all available features to build a model, it is considered that a strategy to pursue the best performance with minimal variables is necessary [38]. This is because it takes a lot of effort and cost to obtain the necessary data in the medical field, and in fact, it is very rare that all the variables necessary for the model are available at the same time. Data mining and machine learning in the medical field have various complex problems ranging from technical problems of data itself to ethical and sociological problems [39]. If these characteristics are not well understood, there is a possibility that the derived knowledge may not be used in clinical practice even if they show high technical performance [40].

Then, the selected variables were used to perform classification analysis with logistic regression, random forest, and SVM algorithms. In addition, the accuracy, precision, recall, F1 score, and Brier score metrics were used to evaluate the performance of the classification algorithms.

In this study, the mortality caused by road traffic accidents was classified into binary categories. However, the sample was highly imbalanced because the number of people who died from traffic accidents accounted for 1.2% of the total number of people who were injured in the accidents. In such cases, where the sample is highly imbalanced, using the original data to perform analyses will result in a distortion of the accuracy of the model. Thus, this study used the SMOTE technique, a synthetic oversampling method, to increase the number of samples of the minority class before performing the analyses.

The performance of the models was evaluated using five evaluation metrics after the four classification algorithms were used to perform classification analysis. As a result, this study found that the evaluation result values ranged from a minimum value of 0.753 to a maximum value of 0.942, which indicates that the performance was above a certain level. In terms of accuracy, which is the most commonly used performance evaluation metric, the models that used the random forest and RBF-SVM algorithms were superior to the models that used the logistic regression and linear SVM algorithms. Additionally, the models that used the random forest and RBF-SVM algorithms were also found to be superior to those that used the logistic regression and linear SVM algorithms according to the following metrics: the precision metric, which measures the ratio of true positives to all the predicted positives, as well as the F1 score, which is the harmonic mean of the precision and recall. Therefore, of the algorithms that classify the mortality related to external injuries caused by road traffic accidents, it can be said that the random forest and RBF-SVM algorithms are superior to the logistic regression and linear SVM algorithms, and that of those two, the RBF-SVM algorithm is the best. It can be deduced that the RBF-SVM algorithm is better than the linear SVM algorithm because the use of multiple variables with diverse characteristics to perform classification analysis, as in this study, causes the dimensions of models to grow complex. Therefore, the performance of a high-dimensional transformed RBF model is superior to that of the SVM model, which uses a two-dimensional linear kernel [36].

Furthermore, it was determined that the random forest technique is a relatively superior method. Random forest is a type of ensemble technique. With the general bagging method, the influence of strong predictors leads to the creation of similar trees. However, with random forests, this method results in decorrelated trees because only the *m* predictors that had been randomly selected from the total set of *p* predictors were taken into consideration to create split criteria. Additionally, only the new random selection of *m* predictors is considered to create the next split. 

In this study, classification analysis was performed after the oversampling method was used to correct the sample data. However, even if the original data were used to evaluate the models before the data were oversampled, the models would have an accuracy of 98% and thus appear to be optimal prediction models. However, when the sample of a study indicated that 98.8% of the sample survived, as it did in this study, and an algorithm was used to predict that the entire sample had survived, the model would still have a faulty accuracy of 98%. Therefore, this study derived a model with greater validity by performing a classification analysis after solving the problem of imbalanced data. This was done to increase the accuracy of mortality predictions for individuals who are injured in road traffic accidents. 

The significance of this study lies in the fact that it demonstrates the limitations of using original data to perform analyses when the data include imbalanced, multidimensional data such as the data on the mortality of traffic accidents. Variables necessary for predicting the mortality due to a road traffic accident should be added to various attributes related to the accident. The attributes include accident time, vehicle speed, and road conditions affected by season or climate. In addition, the severity of the accident, such as the speed limit, the wearing of a seat belt or helmet [41], the change in physiological conditions during a hospital stay, or the severity score of the traumatic injury should be included in order to design a more clear mortality prediction model. However, the limitation of this study was that the dataset used for the analysis had to be conducted to generate a mortality prediction model using public data. Among the limited data items, this study was conducted with an emphasis on the fact that there are not many existing studies related to the prediction of mortality due to road traffic accidents. Through the importance analysis of determining mortality, the important influencing factors were derived, and the study focused on defining the analysis method with the most reasonable performance for unbalanced data through performance comparison evaluation according to the machine learning method. Among the injuries that are mainly caused by traffic accidents, complex factors such as the level of severity, type of complex injury, and underlying disease, can affect the mortality rate. Thus, further research investigating these complex factors is needed. For future research, we have planned to conduct the traffic accident mortality prediction model that can be generalized, including major and additional injuries, treatment information, and underlying diseases, using the analyzed injury severity patterns.

## Figures and Tables

**Figure 1 ijerph-18-05604-f001:**
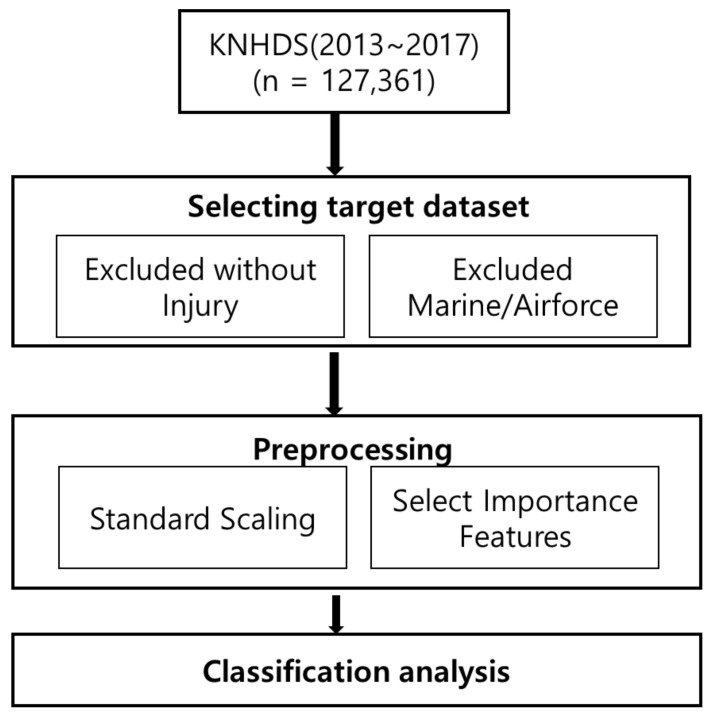
Sample data processing process.

**Figure 2 ijerph-18-05604-f002:**
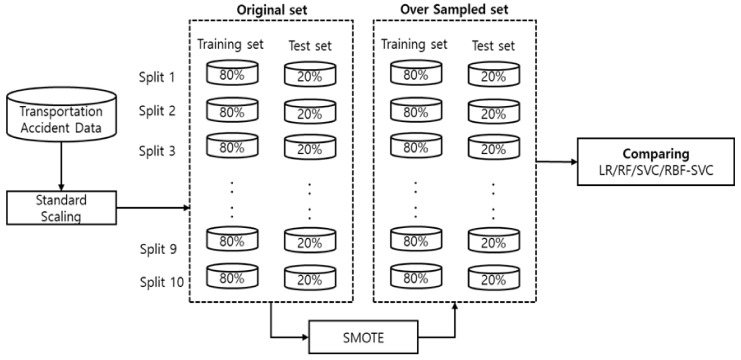
Sample data processing process.

**Figure 3 ijerph-18-05604-f003:**
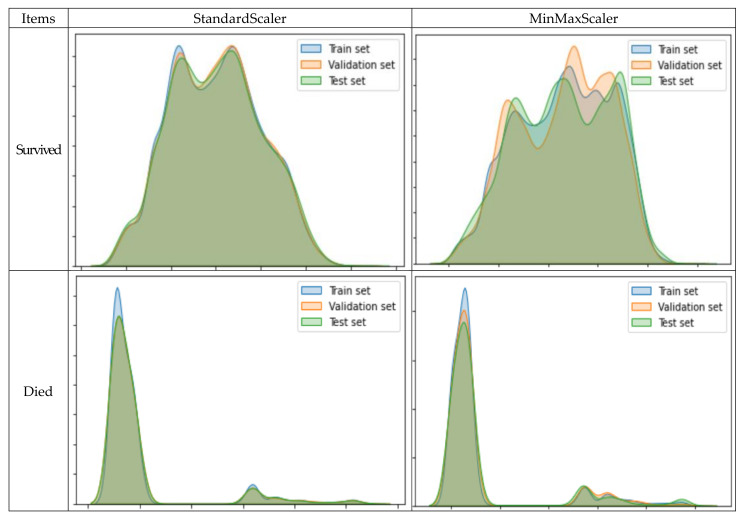
Comparison of standardization methods.

**Figure 4 ijerph-18-05604-f004:**
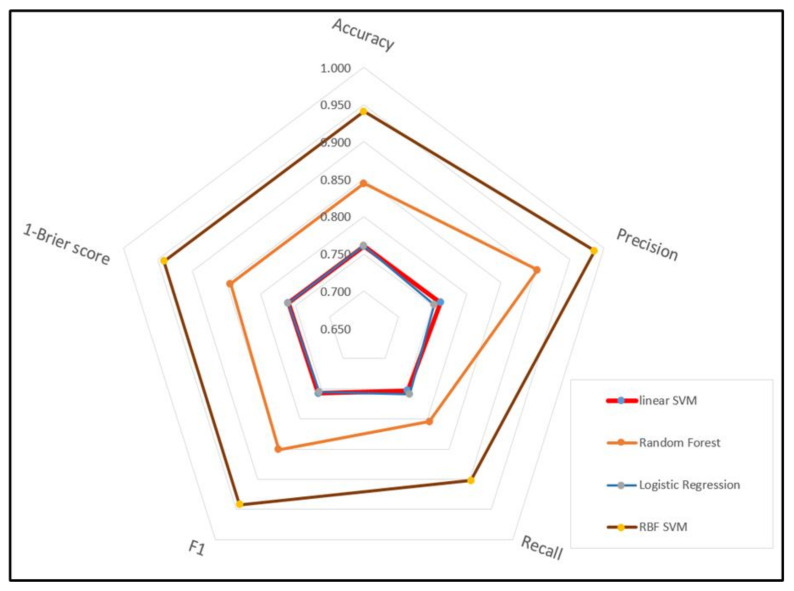
Performance measure among models.

**Figure 5 ijerph-18-05604-f005:**
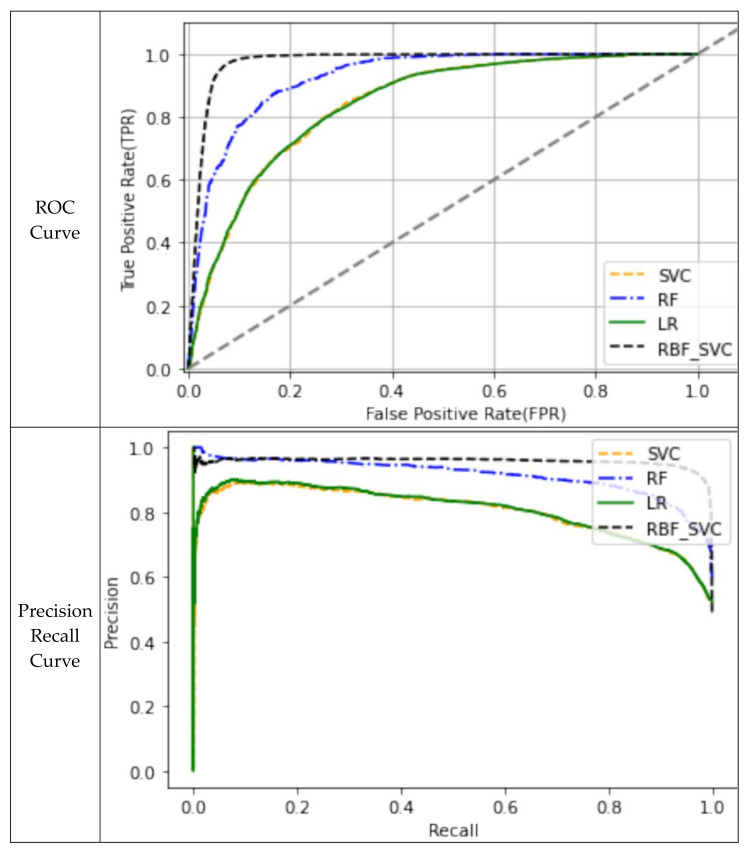
Evaluation Graphs.

**Table 1 ijerph-18-05604-t001:** Frequency analysis.

Items		Freq. (%)	Items		Freq. (%)
Mean age		42.72 (18.401) *	Patterns of principal injury	Superficial injury	9037 (16.3)
Sex	Male	32,936 (59.6)		Open wound	1825 (3.3)
	Female	22,343 (40.4)		Fracture	11,035 (20.0)
Role in accident	Pedestrian	9411 (17.0)		Sprain/dislocation	24,049 (43.5)
	Driver	22,358(40.4)		Nerve injury	289 (0.5)
	Public transit Passenger	8818 (16.0)		Blood vessel injury	87 (0.2)
	Person injured while boarding or exiting vehicle	5699 (10.3)		Internal organ injury **	8147 (14.7)
	Person outside of vehicle	218 (0.4)		Muscle injury	457 (0.8)
	Car passenger	5795 (10.5)		Crush injury	119 (0.2)
	Other Injured person	2980(5.4)		Amputation	34 (0.1)
Primary site of injury	Head	11,050 (20.0)		Other and unspecified injury	200 (0.4)
	Neck	10,523 (19.0)	Type of injury	Multiple site	2481 (4.5)
	Spine	7 (0.0)		Single site	52,726 (95.4)
	Chest	4625 (8.4)		Unspecified	72 (0.1)
	Abdomen/back	11,129 (20.1)	Operation	Yes	11,788 (21.3)
	Shoulder/upper arm	4125 (7.5)		No	43,491 (78.7)
	Forearm	1660 (3.0)	Treatment outcome	Survived	54,609 (98.8)
	Wrist/hand	1669 (3.0)		Died	670 (1.2)
	Hip/thigh	1580 (2.9)			
	Knee/lower leg	5756 (10.4)			
	Ankle/foot	2136 (3.9)			
	Multiple site	933 (1.7)			
	Unknown site	86 (0.1)			

* average (SD), ** Injury of internal organ composite of brain, thoracic and abdominal cavities.

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
