# Peer review of "Comparison of Prediction Models for Mortality Related to Injuries from Road Traffic Accidents after Correcting for Undersampling"

_ijerph, 2021, doi:10.3390/ijerph18115604_

Round 1

Reviewer 1 Report

Summary of paper:

In this manuscript, the authors used KDCA's National Hospital Discharge Survey (National Hospital Discharge Survey) reported five years of data from 2013 to 2017 to compare classification models for predicting road traffic accident mortality. First of all, through importance analysis, the determinants that affect mortality are screened out. Then the selected variables are classified and analyzed by logistic regression, random forest and SVC algorithm. In addition, the accuracy, precision, recall, F1 score and Brier score indicators are used to evaluate the performance of the classification algorithm.

Major comments:

  1. The model of this manuscript lacks generalization ability and needs to be improved.
  2. Whether the SMOTE method is suitable for the use of this data set, this manuscript does not explain the feasibility.
  3. This manuscript only analyzes the results of the model, and does not perform a detailed analysis on the algorithm.
  4. This manuscript is too rough to describe the classification of the algorithm, I cannot judge the true validity of the classification accuracy.
  5. I think the completeness of this manuscript is very good, but the details of each part are not comprehensive.

Minor comments:

  1. Some formulas and diagrams in this manuscript are incorrectly formatted and need to be corrected.
  2. The clarity of the chart needs to be improved.

Author Response

Thank you for your interest and thoughtful comments on our paper, “Comparison of prediction models for mortality related to external injuries from road traffic accidents after correcting for undersampling”, submitted for publication in International Journal of Environmental Research and Public Health. Please find our explanations and answers that may hopefully address the issues commented, which are given in our newly submitted version of the manuscript. The changes made in the manuscript are highlighted in yellow so that they can be easily traced – they will be organized separately at the end as an attachment file. The characters in red refer to the comments made by reviewers.

Reviewer 2 Report

In this paper, the authors dealt with the problem of class imbalance in creating a classification model. The adopted solutions seem efficient but in "Results" are reported metrics that evaluate the global performances of each model. I suggest adding confusion matrices for each model to show how different models perform in identifying the less represented class. 

Author Response

(The authors gave the same response as above.)

Reviewer 3 Report

This study (IJREPH-1191381) has compared the accuracy of four models in predicting mortality caused by road accident injuries, based on data from a national-level survey. The topic is quite important and should be of interests to readers of IJREPH. However, the following issues should be addressed before being published.

  1. The major issue is that the contribution of this study is not clear. Traffic accident or injury models have been intensively investigated during the last few decades. Algorithms such as logistic regression, support vector machine, and other machine learning techniques (e.g., Assi, Rahman, Mansoor, & Ratrout, 2020; Hosseinzadeh, Moeinaddini, & Ghasemzadeh, 2021) have been applied to build the prediction models. What are the innovation points of this study? Although the authors mentioned that they have considered the imbalance problem in the data and tried to solve this problem using the synthetic minority oversampling technique, this innovation alone is not enough to support the publication of the study.
  2. A second major issue is only model evaluation metrics were reported. The authors did not discuss anything about the predictors. For instance, how the characteristics of the patients contributed to the injury risk? Without such results, the implications of this study are not clear.

Some minor issues are listed below.

  1. The first sentence of the abstract. It was said that four models were compared, but the authors only listed three.
  2. Please define RBF first before using this abbreviation.
  3. The abstract should discuss the specific results and implications of this study.
  4. Page 1, what does OECD countries mean?
  5. Some sentences are very confusing. For instance, Line 63 “However, because of the difference in the number of survivors and deceased individuals, it is difficult to use existing methods to develop an effective model for analyzing mortality due to road traffic accidents.” It is not clear what kind of existing methods the authors referred to and why such methods were not suitable for analyzing the data. Another example, line 167, “with superficial injuries classified as minor or deep injuries classified as severe injuries”. Did the authors mean ““with superficial injuries classified as minor AND deep injuries classified as severe injuries”?
  6. The authors did not explain why decision tress, SVC, and random forests were suitable for imbalance data. To me, the above methods were just conventional classification algorithms.
  7. The current form of Table 1 makes it very difficult to understand.
  8. Figure 3 was not cited in the text.
  9. Table 2 is not a standard table.

Author Response

(The authors gave the same response as above.)

Round 2

Reviewer 1 Report

Comments: The author of this manuscript responded to the generalization ability, the details of the model, the calculation results of the model and the classification accuracy, and made corresponding amendments to the tables and formulas. I hope that the author’s answers to my questions can be reflected in the newly revised manuscript. I agree that the manuscript be accepted by this journal.

Author Response

Thank you for your interest and thoughtful comments on our paper, “Comparison of prediction models for mortality related to external injuries from road traffic accidents after correcting for undersampling”, submitted for publication in International Journal of Environmental Research and Public Health. We have organized the paper according to your comments.

Reviewer 3 Report

I would like to thank the authors for considering my comments. Most of my comments have been addressed. However, there are still some formatting issues. For instance, page 2, line 84, SMOTE was not defined; page 14, line 234, “1.2%(670)”; page 9, the formula 1 and 2 are not clear. It is recommended to carefully go through the manuscript and correct all these errors.

Author Response

(The authors gave the same response as above.)
